# OpenReview forum: "In-Context Watermarks for Large Language Models"
_ICLR.cc/2026/Conference — ICLR 2026 Poster_

### Official Review · Reviewer_PAYJ · 2025-10-21

**Soundness:** 2
**Presentation:** 2
**Contribution:** 1
**Rating:** 2
**Confidence:** 5

**Summary:**

This paper explores In-Context Watermarking for LLMs, a kind of approach that embeds watermarks into LLM-generated text solely through prompt engineering. The authors test four ICW strategies operating at different linguistic granularities. The paper evaluates these methods in two settings: Direct Text Stamp and Indirect Prompt Injection. The work positions ICW as a promising model-agnostic watermarking paradigm that empowers third parties to trace AI-generated content without requiring cooperation from model providers.

**Strengths:**

- The authors systematically evaluate trade-offs among LLM requirements, detectability, robustness, and text quality for each strategy, providing valuable insights into the design space of ICW methods.
- The authors acknowledge that ICW effectiveness depends critically on model capabilities, with weaker models essentially unable to follow complex watermarking instructions. The adaptive attack experiments (Table 10) showing partial watermark removal are included transparently. This intellectual honesty strengthens the paper's credibility.
- The Indirect Prompt Injection setting for detecting AI-generated academic reviews represents a timely motivated use case.

**Weaknesses:**

- The paper's primary weakness lies in presenting existing techniques as novel contributions while failing to properly acknowledge prior work. In-context watermarking is not a new concept. The idea of using prompts to embed watermarks has been explored in multiple prior works, yet the authors provide no discussion of this existing literature in their Related Work section. More critically, the four proposed ICW strategies lack originality and are straightforward applications of well-known techniques. Unicode ICW using zero-width characters is an extremely common information hiding technique that has been extensively studied for decades, representing standard steganography practice rather than a novel contribution. The Lexical ICW directly borrows the green/red-list concept from KGW, the only difference is implementation via prompts rather than logits manipulation. Acrostic steganography has centuries of history, and recent work has already demonstrated using ChatGPT for acrostic-based information hiding (https://daniellerch.me/stego/text/chatgpt-en/), yet this precedent is not mentioned. Furthermore, the paper provides no meaningful insights beyond these straightforward applications. The contribution essentially reduces to showing "you can ask LLMs to follow watermarking instructions via prompts", which is a trivial observation given modern LLMs' instruction-following capabilities.

- Even setting aside novelty concerns, the proposed methods demonstrate fundamental practical limitations that severely constrain their real-world applicability. As the authors acknowledge, these heuristic approaches significantly compromise output text quality. The methods are also fragile against simple attacks: even without knowing the specific watermarking scheme, adversaries can substantially remove watermarks through text rewriting. This fundamental vulnerability undermines the practical utility of ICW. Additionally, the approach exhibits strong model capability dependency, only working with the most advanced models like GPT-o3-mini.

- The proposed ICW methods can be understood as straightforward applications of well-established concepts without meaningful innovation. The IPI setting is essentially a benign variant of indirect prompt injection attacks, the novelty of "using this for watermarking instead of attacks" is incremental at best. Similarly, instructing models to embed hidden patterns via prompts is conceptually identical to in-context backdoor insertion, yet the paper does not acknowledge or discuss this connection.

- The ICW methods suffer from fundamental compatibility issues with diverse writing scenarios. For instance, Acrostics ICW, which constrains sentence-initial letters to follow a predetermined sequence, is unsuitable for many real-world applications such as technical documentation and code comments.

- The paper claims to be the "first" to explore ICW, but provides no systematic literature review to support this claim.

**Questions:**

1. Please clarify the novelty of the four proposed heuristic ICW methods and explain how they differ from existing approaches.

2. Please compare your methods with a broader range of in-context watermarking approaches.

3. Given that attackers can easily remove watermarks without knowing the specific algorithm, how can these methods be applied in practice?

4. In writing scenarios that prohibit the structural or stylistic patterns required by your ICW methods, how can your approach be applied?

---

> ### Author Response · Authors · 2025-11-14
>
> Dear reviewer PAYJ,
>
> Thank you very much for taking the time to read our paper and for providing thoughtful and constructive feedback. We truly appreciate your insights. We are carefully reviewing all of your comments and are working to address the concerns you raised.
>
> In your review, we noticed that you mentioned that “in-context watermarking is not a new concept” and that “the idea of using prompts to embed watermarks has been explored in multiple prior works”. **To ensure that we fully understand the prior related works, accurately reflect them in our revised manuscript, and better address your concerns, would you be willing to point us to the specific works you had in mind?** We would sincerely appreciate your guidance, and **we will make sure to carefully discuss and include any relevant references in the next version of the manuscript**.
>
> Thank you again for your time and for helping us improve our work.
>
> Best regards,
>
> Authors of Paper 7815

---

> > ### Comment · Reviewer_PAYJ · 2025-11-14
> >
> > Dear Authors,
> >
> > Thank you for your response. I appreciate your willingness to engage with the feedback and improve your manuscript.
> >
> > Regarding in-context watermarking as a concept, while in-context learning itself is indeed a fundamental and well-established technique, let me point you to some relevant works that have explored watermarking in this context:
> >
> > 1. https://aclanthology.org/2025.acl-long.145.pdf This work discusses prompt engineering for watermarking, noting: "Prompt Engineering. Another alternative is to use the inherent capabilities of LLMs by designing a system prompt that specifies the desired features (Zhou et al., 2023). For example, it is possible to tell contemporary LLMs to use only certain letters in their sentences (OpenAI ChatGPT, 2024)." While they dedicate only a brief section to this approach, since it represents a relatively straightforward and naive solution, it demonstrates that the idea has been previously explored.
> >
> > 2. https://arxiv.org/pdf/2411.05091 This paper explicitly introduces "a prompt-guided watermarking framework that operates entirely at the input level and requires no access to model parameters or decoding logits."
> >
> > 3. In-context backdoor attacks (https://arxiv.org/pdf/2307.14692) From a few years ago, this work explored in-context backdoors. In-context watermarking can be viewed as essentially the same technical approach, just framed with a different narrative.
> >
> > I hope these references help you better position your work within the existing literature and revise your manuscript accordingly.
> >
> > Additionally, I'd encourage you to consider: what novel and unique methodology can you propose for in-context watermarking? Rather than relying on approaches that are relatively straightforward and could be conceived by anyone familiar with the field, I believe your paper would benefit significantly from introducing more innovative technical contributions.
> >
> > I wish you the best in revising your work.
> >
> > Best regards,
> > Reviewer

---

> ### Author Response · Authors · 2025-11-20
>
> We sincerely thank you for carefully reading our paper and for your constructive comments. We have carefully considered your valuable questions and provided detailed responses below.
>
> ### **W1&W3&W5&Q1&Q2. Claim the contribution of this work and discuss prior works.**
>
> 1. We appreciate your careful attention to prior works in text steganography. We fully agree that individual linguistic mechanisms such as zero-width Unicode characters, lexical biasing, or acrostics are longstanding steganographic primitives.
>
> **However, our work’s novelty does not lie in inventing new low-level steganographic forms, but in formulating and empirically validating the in-context watermarking paradigm as a rigorous, model-agnostic setting for in-process watermarking.**  Moreover, we want to mention that most current LLM watermarking methods are based on those low-level steganographic methods and unique linguistic characteristics.
>
> Specifically, we highlight our contribution as follows. Our work:
>
> - Provides the **first formal problem formulation of ICW**, including watermark embedding, detector design, hypothesis testing-based evaluation, and false-alarm analysis.
>
> - Offers **systematically evaluate the proposed ICW strategies** across advanced LLMs and both DTS and IPI settings, including detectability, robustness, text quality, potential attacks, and comparison with baselines using metrics such as ROC-ARC, TPR at low FPR, etc.
>
> - Shows that watermark detectability and robustness scale with instruction-following capabilities, a **novel empirical finding** suggesting that advances in LLMs can make watermarking easier and inform future LLM alignment and policy enforcement.
>
> - Introduces the indirect prompt injection (IPI) setting, where watermarking can be embedded covertly in long-context documents (e.g., peer review use case), **a scenario which is not studied in any prior work.** **It provides a broader application scenario of ICWs.**

---

> ### Author Response · Authors · 2025-11-20
>
> ### **W1&W3&W5&Q1&Q2. Claim the contribution of this work and discuss prior works.**
>
> 2. We thank the reviewer for sharing these relevant references. We have carefully studied each of them. **We respectfully do not fully agree with the viewpoint that ICW has been extensively explored in previous works.** We provide the following discussion for comparison.
>
> - [1] **just mentioned** that the prompt could be a way to add a watermark to LLMs in the related work section. However, **it doesn’t propose or evaluate any ICW-related methods or experiments**. Their method is based on logits perturbation and cannot be directly applied in our model-agnostic setting.
>
> - While [2], a concurrent work, proposes a prompt-based watermarking method, **our study differs substantially in formulation and methodology.** **(a)** [2] trains a classifier as a detector and reports only detection accuracy, without providing any statistical guarantees. In contrast, our work formulates watermark detection as a hypothesis testing problem, employs statistical detectors with false-alarm guarantees, and evaluates performance using ROC-AUC and true positive rates at low false positive rates. **(b)** We conduct an extensive evaluation of the proposed methods, including detection performance (ROC-AUC), text quality, robustness, etc. In contrast, [2] mainly evaluates the detection accuracy without text quality and robustness. **(c)** In addition to the DTS setting, we propose the IPI setting, detailing how watermark delivery occurs in realistic, high-stakes workflows, which have never been explored in previous works, and provide insight into broader applications.
>
> - **[3] is a backdoor attack method, which is completely different from our work** as follows: **(a)** Objective: [3] targets to misbehavior of LLM (e.g., force a chosen label on classification when a trigger appears), while our ICW focuses on text provenance/attribution (e.g., embed detectable yet unobtrusive signals in model outputs to identify AI-assisted text). **(b)** Threat model & access: [3] **inserts a backdoor by fine-tuning the LLMs, which is similar to traditional backdoor methods.** However, our **ICW is completely black-box (prompt-only and model-agnostic), which is different from previous LLM watermarking methods** (perturb the decoding process). **Even though [3] mentioned in-context learning, it studies a completely different research problem.**
>
> - [4] is an **online blog that only presents simple demonstrations of text steganography using ChatGPT** with acrostics. It does not provide a formal problem formulation or any systematic or quantitative evaluation, in contrast to our work (see our contributions above).
>
> We thank the reviewer for pointing out those prior works [1-4]. We also find some steganography-related works [5-8]. **We assure that we will include all those works and discussions in the revised manuscript.**
>
> Lastly, we would like to emphasize that **conceiving the right primitives, adapting them to this new setting, formulating the problem, and carrying out extensive evaluation is itself a non-trivial and intellectually interesting contribution, as also recognized by Reviewers mcZt, Asf9, and whPx.**
>
> [1] https://aclanthology.org/2025.acl-long.145.pdf
>
> [2] https://arxiv.org/pdf/2411.05091
>
> [3] https://arxiv.org/pdf/2307.14692
>
> [4] https://daniellerch.me/stego/text/chatgpt-en/
>
> [5] Meral, Hasan Mesut, Bülent Sankur, A. Sumru Özsoy, Tunga Güngör, and Emre Sevinç. "Natural language watermarking via morphosyntactic alterations." Computer Speech & Language 23, no. 1 (2009): 107-125.
>
> [6] Topkara, Mercan, Umut Topkara, and Mikhail J. Atallah. "Words are not enough: sentence level natural language watermarking." In Proceedings of the 4th ACM international workshop on Contents protection and security, pp. 37-46. 2006.
>
> [7] Por, Lip Yee, KokSheik Wong, and Kok Onn Chee. "UniSpaCh: A text-based data hiding method using Unicode space characters." Journal of Systems and Software 85, no. 5 (2012): 1075-1082.
>
> [8] Brassil, Jack T., Steven Low, Nicholas F. Maxemchuk, and Lawrence O'Gorman. "Electronic marking and identification techniques to discourage document copying." IEEE Journal on Selected Areas in Communications 13, no. 8 (1995): 1495-1504.

---

> ### Author Response · Authors · 2025-11-20
>
> ### **W2&Q3. Practicability of real-world application, including text quality, robustness, and model dependency.**
>
> We sincerely thank the reviewer for pointing out these concerns. **We believe the reviewer may have some misunderstandings regarding our paper.** Therefore, we would like to clarify them as follows.
>
> - **We never claim that our methods ‘significantly compromise output text quality’.** As shown in Table 3 of our manuscript, we evaluate the relevance, clarity, and quality of the watermarked responses using LLM-as-a-Judge. The results show that **the overall text quality of our ICW methods is comparable to human and un-watermarked text, and significantly better than other baselines.**
>
> - The reviewer claims that ‘adversaries can substantially remove watermarks through text rewriting’. For this concern, **we explicitly evaluate the robustness of our ICW methods under a full paraphrasing attack**, where an LLM is instructed to completely rewrite each watermarked text. As shown in Figure 3 of our manuscript, the detection performance of the Initials, Lexical, and Acrostics ICW methods remains high, achieving ROC-AUC scores of 0.887, 0.924, and 0.922, respectively, substantially outperforming all baseline methods. **These results demonstrate that our ICW approaches preserve strong detectability even after rewriting.**
>
> - **It is an important empirical finding of our work that the effectiveness of ICW scales with the underlying model’s capability, which has never been explored in previous work.** Moreover, **we view this not as a limitation, but as evidence of ICW’s forward-looking potential.** As LLMs continue to advance in instruction-following and long-context understanding, ICW naturally becomes more reliable and robust. Notably, our experiments already demonstrate strong performance on current advanced models such as GPT-o3-mini, suggesting that ICW will become increasingly practical and powerful as model capabilities improve further.
>
> ### **W4. Compatibility of ICWs with real-world writing scenarios.**
>
> We do not intend to propose a single method that can be applied universally across all application scenarios, which is difficult even for traditional LLM watermarking approaches that rely on modifying logits.
>
> Instead, we believe that **Different use cases naturally require different trade-offs between detectability, robustness, and text naturalness. This perspective directly motivates the different watermarking algorithms we proposed in the paper.**
>
> Importantly, unlike conventional watermarking methods that must be deployed by LLM providers, ICW can be flexibly implemented by third-party stakeholders according to their specific goals and contexts. For example, in constrained formats such as technical documentation or code comments, where sentence structure is rigid, Unicode, Initials, or Lexical ICWs are appropriate. By contrast, for long-form prose (e.g., blogs, reviews, literature), Acrostics ICW is well-suited and can deliver high robustness with minimal quality impact.

---

> ### Author Response · Authors · 2025-11-25
> **Follow-up discussion**
>
> Dear reviewer PAYJ,
>
> We sincerely appreciate your time and effort in reviewing our submission and providing valuable suggestions. While we hope to have addressed your concerns adequately, we understand there may still be areas requiring further clarification or discussion. We are fully prepared to address your outstanding issues. Should our responses have successfully addressed all your questions, we would be deeply grateful if you could consider enhancing the score to further support our submission. Thank you very much for your thoughtful review.
>
> Best Regards,
>
> Authors of Paper 7815

---

> > ### Comment · Reviewer_PAYJ · 2025-11-26
> >
> > I appreciate your effort in providing a rebuttal. However, as we have differing views on the novelty and contribution of the paper, and your explanations have not persuaded me, I have chosen to maintain my original score.

---

### Official Review · Reviewer_Asf9 · 2025-10-30

**Soundness:** 3
**Presentation:** 3
**Contribution:** 3
**Rating:** 8
**Confidence:** 4

**Summary:**

This paper introduces ICWs, a novel approach to watermark LLMs generated text without modifying the decoding process. Instead, it embeds watermarks through prompt engineering, leveraging LLMs’ in-context learning and instruction-following capabilities. Specifically, this paper explores four ICW strategies, each with different embedding and detection schemes. These strategies are evaluated across two settings: DTS and IPI. Comprehensive experiments demonstrate the effectiveness of the proposed methods, particularly with more capable LLMs.

**Strengths:**

1.	The concept of embedding watermarks through prompt engineering instead of manipulating the decoding process is interesting and novel, especially under the black-box setting. In this scenario, the control of the LLM watermark is not limited to the LLM owner, which provides a broader application scenario.
2.	This paper evaluates the ICWs on two real-world applications, DTS and IPI, demonstrating the practicability.
3.	This paper conducts an extensive experimental evaluation, demonstrating the effectiveness of the proposed methods.

**Weaknesses:**

1.	The proposed ICWs rely on the instruction following capability of LLMs. Authors should evaluate whether the watermarking instruction can still be reliably followed after a multi-turn conversation.
2.	The current design of the watermarking instruction is intuitive. I am interested in whether the watermarking instruction could be further optimized using RL or some prompt tuning methods (just for discussion).
3.	Authors should add some discussion about the reason that Lexical and Acrostics ICWs show lower detection performance under the deletion attack.

**Questions:**

1. This paper introduces one method, e.g., white text, to covertly embed the watermarking instruction into the confidential file. Can you introduce more potential methods to covertly embed the watermarking instruction?
2. I am interested in the robustness performance of ICW under the copy-paste attack.
3. How is the detection threshold \(\eta\) in Eq. (H1) of Sec. 3.1 chosen in practice?
4. Is it dataset-specific, or does a universal threshold generalize across domains?
5. What additional computational cost (token length, inference latency) does each ICW introduce relative to normal prompting?

---

> ### Author Response · Authors · 2025-11-20
>
> We sincerely thank you for the positive and encouraging comments. We are glad that you found our idea interesting and recognized its novelty and practical value in enabling watermarking under the black-box setting. For your valuable reviews, we provide our detailed responses below.
>
> ### **W1. Evaluate the detectability of ICWs after a multi-turn conversation.**
>
> We performed a long, multi-turn conversation (10 turns) after setting the system prompt to evaluate whether the watermarking instruction continues to be followed with an extended context in the DTS setting. The results, presented in the following table, demonstrate that our ICW methods maintain strong detection performance even with 10-turn conversations.
>
> | Methods       | ROC-AUC | TPR@1%FPR | TPR@10%FPR |
> |---------------|---------|-----------|------------|
> | Unicode ICW   |  1.000  |   1.000   |    1.000   |
> | Initials ICW  |  0.999  |   0.988   |    1.000   |
> | Lexical ICW   |  0.995  |   0.926   |    0.995   |
> | Acrostics ICW |  1.000  |   1.000   |    1.000   |
>
> ### **W2. Discuss whether the watermarking instruction could be further optimized using RL or some prompt tuning methods.**
>
> Thank you for the insightful suggestion. We agree that optimizing the watermarking instructions is a very promising direction for further improving the performance of ICWs. In particular, recent prompt optimization methods such as TextGrad [1] provide a strong starting point for this line of research. We plan to explore this direction in future work to automatically refine watermarking prompts.
>
> [1] Yuksekgonul, Mert, Federico Bianchi, Joseph Boen, Sheng Liu, Zhi Huang, Carlos Guestrin, and James Zou. "Textgrad: Automatic" differentiation" via text." arXiv preprint arXiv:2406.07496 (2024).
>
> ### **W3. Discussion about the reason that Lexical and Acrostics ICWs show lower detection performance under the deletion attack.**
>
> We thank the reviewer for the suggestion. The Lexical and Acrostics ICWs exhibit slightly lower robustness to random deletion attacks. This is primarily because these methods utilize a smaller set of words or letters compared to the Initials ICW. Specifically, the Lexical ICW only considers adjectives, adverbs, and verbs, while the Acrostics ICW relies solely on the first letter of each sentence. We will add this discussion to the revised manuscript to make it clearer.
>
>
> ### **Q1. More potential methods to covertly embed the watermarking instruction.**
>
> Thank you for your good suggestion. There are two more methods to covertly embed the watermarking information in the PDF file. (1) PDF font-embedding (cmap remapping): embed a custom font so that the rendered glyphs appear innocuous to humans while the underlying character codes spell the instruction that LLM text parsers read. This approach is visually stealthy and adds negligible token/latency overhead. (2) Different-language micro-injection: insert a brief, low-salience note in a widely supported secondary language (e.g., small-font French) in a footnote/caption. Humans often skip or cannot quickly interpret it, yet modern LLMs reliably parse it, even after OCR, making it a practical fallback where font tricks fail.
>
> ### **Q2. Robustness performance of ICW under the copy-paste attack.**
>
> We conducted experiments to evaluate the robustness of our ICW methods against a copy-paste attack. Specifically, we copy the watermarked text, insert it into another text of similar length, and then evaluate detection performance on this mixed sample containing both watermarked and unwatermarked content. The results, presented in the following table, demonstrate that our proposed ICW methods maintain strong robustness under copy-paste attack.
>
> | Methods       | ROC-AUC | TPR@1%FPR | TPR@10%FPR |
> |---------------|---------|-----------|------------|
> | Unicode ICW   |  1.000  |   1.000   |    1.000   |
> | Initials ICW  |  0.990  |   0.922   |    0.972   |
> | Lexical ICW   |  0.953  |   0.606   |    0.868   |
> | Acrostics ICW |  1.000  |   1.000   |    1.000   |
>
> ### **Q3. How is the detection threshold ($\eta$) in Eq. (H1) of Sec. 3.1 chosen in practice?**
>
> The detection threshold can be selected based on the estimated false alarm rate. We provide a theoretical guarantee on controlling the false alarm rate for Initials and Lexical ICWs in Appendix B. In this way, we could select the detection threshold based on the desired false alarm rate. For Acrostics ICW, we compute a Levenshtein distance based z-statistic to compare the sequence of sentence initials with the secret key, and apply a preset significance level to control false alarms.

---

> ### Author Response · Authors · 2025-11-20
>
> ### **Q4. Is it dataset-specific, or does a universal threshold generalize across domains?**
>
> Thank you for raising this point. Our detector thresholds are chosen by directly controlling the false-positive rate to be below a pre-defined level. Under the null hypothesis $H_0$ (“the text is not watermarked”), our key structural assumption is that the generated text is independent of the green/red list (or initial letters) derived from the secret key. This assumption does not depend on any particular dataset or writing domain. Consequently, the false-positive rate does not change under domain shift, and the same threshold is universal across domains. Changes in writing style or topic may affect the detection power (TPR), but they do not increase the false-positive rate. This design ensures that ICW detection remains stable and reliable across diverse text domains.
>
> ### **Q5. What additional computational cost (token length, inference latency) does each ICW introduce relative to normal prompting?**
>
> It depends on the specific ICW strategy. Unicode, Initials, and Acrostics ICWs add only a small prompt overhead (the watermarking instruction/key), so they slightly increase context length but do not significantly impact the output length, hence the latency impact is negligible. For Lexical ICW, the cost depends on the size of the green/red list: a longer list increases input tokens and therefore can modestly increase encoding latency and context usage. In practice, this is easily controlled by (1) keeping the instruction concise, (2) using topic-aware pruning to activate only a small set of relevant terms per document (IPI setting). With these settings, the additional computation remains small and comparable to standard prompting.

---

> > ### Comment · Reviewer_Asf9 · 2025-11-24
> > **Reviwer's response**
> >
> > I appreciate the authors’ thorough rebuttal. Their responses effectively address several of my concerns and enhance the paper’s practical relevance. I will consider this rebuttal when making my final recommendation.

---

> > > ### Author Response · Authors · 2025-11-24
> > >
> > > Thank you very much for your thoughtful follow-up and for acknowledging our rebuttal. We sincerely appreciate the time and effort you put into reviewing our work. Your comments and suggestions have been extremely valuable in improving the clarity, practical relevance, and overall quality of the paper. We are grateful for your positive rating and for considering our response in your final recommendation!
> > >
> > > Best Regards,
> > >
> > > Authors of Paper 7815

---

### Official Review · Reviewer_whPx · 2025-10-31

**Soundness:** 3
**Presentation:** 4
**Contribution:** 4
**Rating:** 6
**Confidence:** 4

**Summary:**

The paper proposes a family of In-Context watermarking schemes that require no access to model internals, relying instead on watermarking instructions to induce detectable patterns in LLM outputs. Experiments on proprietary models (GPT-4o-mini and GPT-o3-mini) show that detection improves with model capability in both DTS and IPI settings. The paper reports robustness against edits/paraphrasing and small quality costs as judged by an LLM-as-a-judge.

**Strengths:**

The work formulates ICW embedding/detection without needing logits/decoding, filling a gap between post-hoc detection and in-process watermarking.

The four ICWs span character/word/sentence levels and include concrete detection rules.

Results provide interesting empirical insight, showing the relationship between LLM instruction-following capacity and ICWs' effectiveness.

**Weaknesses:**

Covertly modifying review manuscripts with hidden instructions (even for integrity) may raise consent or transparency issues. Authors should discuss this point.

The detection performance of the proposed method is evaluated only on two LLMs. At least one more LLM is suggested to further demonstrate the effectiveness of the proposed methods.

**Questions:**

In the IPI setting, what safeguards prevent false accusations (e.g., reviewers unintentionally triggering a watermark through unrelated text patterns)?

How does detection performance change as the length of the output increases?

How sensitive are the Initials/Lexical ICWs to domain shifts?

If watermarks are embedded using one model family and text is later post-edited by another, how do detectors fare?

---

> ### Author Response · Authors · 2025-11-20
>
> We sincerely appreciate your recognition of our ICW formulation and empirical insight. We are grateful for your constructive feedback and provide our detailed responses below.
>
> ### **W1. Discuss the consent or transparency issues that may arise from covertly modifying review manuscripts with hidden instructions.**
>
> We sincerely thank the reviewer for raising this important point. We fully agree that covertly embedding watermarking instructions in review manuscripts requires careful consideration of transparency and consent.
>
> In this work, we therefore emphasize that the implementation of ICW should be carried out by authorized stakeholders, for example, conference organizers in the IPI setting, whose incentives are aligned with maintaining the integrity and fairness of the peer-review process. This design choice reflects a central principle of ICW: it empowers legitimate third parties, those who already hold institutional responsibility for content verification or accountability, to perform watermarking without relying on model providers. Traditional watermarking approaches often assume cooperation from model owners, who may lack incentive or control in specific applications. In contrast, ICW enables independent stakeholders, such as academic conferences, publishers, or auditing bodies, to embed and later verify provenance signals through prompt-level interactions only. By entrusting ICW implementation to such authorized entities, the watermarking process remains goal-aligned, policy-governed, and auditable. These entities are best positioned to define appropriate usage policies, ensure compliance with ethical standards, and maintain transparency with affected participants. In this way, ICW serves as a practical and accountable tool for reinforcing integrity in institutional workflows, rather than a mechanism for unsanctioned content manipulation.
>
> ### **W2. Evaluate detection performance of the proposed methods on one more LLM.**
>
> To further demonstrate the effectiveness of our methods, we evaluated our ICW techniques on an additional LLM, gpt-o4-mini. As shown in the following table, our methods consistently achieve high detection performance, which demonstrates the effectiveness of our proposed methods over advanced models.
>
> | Methods       | ROC-AUC | TPR@1%FPR | TPR@10%FPR |
> |---------------|---------|-----------|------------|
> | Initials ICW  |  0.997  |   0.994   |    0.998   |
> | Lexical ICW   |  0.996  |   0.943   |    0.995   |
> | Acrostics ICW |  1.000  |   1.000   |    1.000   |
>
> ### **Q1. In the IPI setting, what safeguards prevent false accusations?**
>
> We provide a theoretical guarantee on controlling the false alarm rate for Initials and Lexical ICWs in Appendix B. In this way, we could select the detection threshold based on the desired false alarm rate. For Acrostics ICW, we compute a Levenshtein distance based z-statistic to compare the sequence of sentence initials with the secret key, and apply a preset significance level to control false positives.
>
> ### **Q2. How does detection performance change as the length of the output increases?**
>
> To further investigate the effects of context and output length on ICW detection performance, we conducted two ablation studies. First, we performed a long, multi-turn conversation (10 turns) after setting the system prompt to evaluate whether the watermarking instruction continues to be followed with an extended context in the DTS setting. Second, we increased the output length from 300 to 1000 words to assess the impact of output length on watermarking effectiveness. All experiments were performed on gpt-o3-mini. The results, presented in the following two tables, demonstrate that our ICW methods maintain strong detection performance even with longer contexts and outputs.
>
> Table 1: Detection performance of different ICW methods for different context lengths.
> | Methods       | ROC-AUC | TPR@1%FPR | TPR@10%FPR |
> |---------------|---------|-----------|------------|
> | Unicode ICW   |  1.000  |   1.000   |    1.000   |
> | Initials ICW  |  0.999  |   0.988   |    1.000   |
> | Lexical ICW   |  0.995  |   0.926   |    0.995   |
> | Acrostics ICW |  1.000  |   1.000   |    1.000   |
>
> Table 2: Detection performance of different ICW methods for different output lengths.
> | Methods       | ROC-AUC | TPR@1%FPR | TPR@10%FPR |
> |---------------|---------|-----------|------------|
> | Unicode ICW   |  1.000  |   1.000   |    1.000   |
> | Initials ICW  |  0.998  |   0.985   |    0.999   |
> | Lexical ICW   |  0.996  |   0.935   |    0.995   |
> | Acrostics ICW |  1.000  |   1.000   |    1.000   |

---

> ### Author Response · Authors · 2025-11-20
>
> ### **Q3. How sensitive are the Initials/Lexical ICWs to domain shifts?**
>
> Thank you for raising this point. Our detector thresholds are chosen by directly controlling the false-positive rate to be below a pre-defined level. Under the null hypothesis $H_0$ (“the text is not watermarked”), our key structural assumption is that the generated text is independent of the green/red list (or initial letters) derived from the secret key. This assumption does not depend on any particular dataset or writing domain. Consequently, the false-positive rate does not change under domain shift, and the same threshold is universal across domains. Changes in writing style or topic may affect the detection power (TPR), but they do not increase the false-positive rate. This design ensures that ICW detection remains stable and reliable across diverse text domains.
>
> ### **Q4. If watermarks are embedded using one model family and text is later post-edited by another, how do detectors fare?**
>
> This is an important aspect of robustness. To evaluate it, we use an LLM (e.g., GPT-4o-mini) to fully paraphrase the watermarked texts and then measure watermark detectability after paraphrasing. The results, shown in the table below, demonstrate that the Initials, Lexical, and Acrostics ICW methods maintain high detection performance even after this paraphrasing attack.
>
>
> | Methods       | ROC-AUC | TPR@1%FPR | TPR@10%FPR |
> |---------------|---------|-----------|------------|
> | Unicode ICW   |  0.500  |   0.010   |    0.100   |
> | Initials ICW  |  0.887  |   0.218   |    0.678   |
> | Lexical ICW   |  0.924  |   0.434   |    0.746   |
> | Acrostics ICW |  0.922  |   0.534   |    0.788   |

---

### Official Review · Reviewer_mcZt · 2025-11-01

**Soundness:** 3
**Presentation:** 3
**Contribution:** 3
**Rating:** 6
**Confidence:** 3

**Summary:**

The paper introduces a new type of LLM watermarking that does not require access to the decoding process. The main idea is to embed a watermarking signal directly into the model's output through prompt modification. The authors propose two ways to do this. The first is by giving the LLM direct instructions in the system prompt, such as asking it to start words with specific letters or follow a detectable pattern. The second is by inserting watermarking instructions into the main body of the input, for example in a PDF that could be used by dishonest reviewers to automatically review papers.

**Strengths:**

- I really like the main idea of the paper. It opens a new and relevant direction in LLM watermarking, with interesting applications such as detecting dishonest reviewers.

- The paper is clearly written and easy to follow.

**Weaknesses:**

- The paper lacks an evaluation of the true positive rate at very low false positive rates. For the method to be practical (for example, in detecting dishonest reviewers), it needs to perform well at very low FPRs (e.g., 0.1%).

- It would also be helpful to include a human evaluation to support the results from the LLM-as-a-judge or perplexity based evaluation, although I do not think this is required for acceptance.

**Questions:**

I think the users could notice (at least for some watermarking strategies) the watermarking patterns either by direct / visual inspection or by asking another LLM to detect them. How would you defend against the possibility that users identify and remove the watermarking patterns?

---

> ### Author Response · Authors · 2025-11-20
>
> We sincerely thank you for the constructive and positive comments. We are pleased that you found our idea interesting and recognized its potential to open a promising new direction in LLM watermarking research. We highly value your review and have carefully discussed your comments in the following responses.
>
> ### **W1. Evaluate the true positive rate at a very low false positive rate (e.g., 0.1%).**
>
> Thank you for this valuable suggestion. To further assess the practicality of our approach, we extended our evaluation to include true positive rates at an extremely low false positive rate of 0.1% (TPR@ 0.1%FPR) in both DTS and IPI settings. The results are presented in the table below. These results demonstrate that ICW remains effective even under highly conservative false-positive thresholds, supporting its practicality for real-world use cases such as detecting AI-generated reviews.
>
>
> | Setting     |    Methods    | ROC-AUC | TPR@ 0.1%FPR | TPR@1 %FPR | TPR@1 %FPR |
> |-------------|:-------------:|:-------:|:-----------:|:----------:|:----------:|
> | DTS setting | Unicode ICW   |  1.000  |    1.000    |    1.000   |    1.000   |
> |             | Initials ICW  |  0.999  |    0.962    |    0.990   |    0.998   |
> |             | Lexical ICW   |  0.995  |    0.880    |    0.930   |    0.994   |
> |             | Acrostics ICW |  1.000  |    1.000    |    1.000   |    1.000   |
> | IPI setting | Unicode ICW   |  1.000  |    1.000    |    1.000   |    1.000   |
> |             | Initials ICW  |  0.997  |    0.840    |    0.910   |    0.998   |
> |             | Lexical ICW   |  0.997  |    0.958    |    0.974   |    0.989   |
> |             | Acrostics ICW |  0.997  |    0.946    |    0.982   |    0.998   |
>
>
> ### **W2. Include a human evaluation to support the results from the LLM-as-a-judge or perplexity based evaluation.**
>
> Thank you for your good suggestion. We fully agree with you that a complementary human evaluation could further support the results from the LLM-as-a-judge or perplexity-based evaluation. Due to time and resource constraints, we were unable to include a full-scale human study in this submission, but we plan to incorporate one in the future to further validate our current evaluation. Moreover, some existing works [1-4] have demonstrated the effectiveness of LLM-based evaluation and its good consistency with human evaluation on text quality. We will revise our manuscript to include these discussions in the updated version.
>
> [1] Chen, Yi, Rui Wang, Haiyun Jiang, Shuming Shi, and Ruifeng Xu. "Exploring the use of large language models for reference-free text quality evaluation: An empirical study." arXiv preprint arXiv:2304.00723 (2023).
>
> [2] Zheng, Lianmin, Wei-Lin Chiang, Ying Sheng, Siyuan Zhuang, Zhanghao Wu, Yonghao Zhuang, Zi Lin et al. "Judging llm-as-a-judge with mt-bench and chatbot arena." Advances in neural information processing systems 36 (2023): 46595-46623.
>
> [3] Kocmi, Tom, and Christian Federmann. "Large language models are state-of-the-art evaluators of translation quality." arXiv preprint arXiv:2302.14520 (2023).
>
> [4] Ji, Yunjie, Yan Gong, Yiping Peng, Chao Ni, Peiyan Sun, Dongyu Pan, Baochang Ma, and Xiangang Li. "Exploring chatgpt's ability to rank content: A preliminary study on consistency with human preferences." arXiv preprint arXiv:2303.07610 (2023).

---

> ### Author Response · Authors · 2025-11-20
>
> ### **Q1. How would you defend against the possibility that users identify and remove the watermarking patterns?**
>
> We sincerely thank you for this sharp and insightful observation. Indeed, user awareness and potential removal of watermarking patterns are important practical considerations for any watermarking approach.
>
> To address this concern, we conducted an adaptive attack on our proposed ICW methods. Specifically, we first prompt an LLM to assess whether a given text is suspected of being watermarked, and then instruct it to attempt to remove the watermark by paraphrasing the text. This attack serves two purposes: first, to explore whether watermarked text can be easily identified as such; and second, if identified, to assess whether the specific watermarking scheme can be discerned and successfully removed. The details of this attack can be found in Appendix D.2 of the manuscript. The results are presented in the following table. For most ICW methods, such as Unicode, Lexical, and Acrostics ICWs, it is difficult for LLMs to accurately identify which texts are watermarked. The Initials ICW exhibits a higher identification rate, indicating that its pattern is somewhat easier to suspect. However, even in this case, the Initials ICW maintains a high ROC-AUC of 0.893 after the removal attack, demonstrating that correctly identifying the presence of a watermark does not necessarily enable successful removal. We will revise our manuscript to further highlight this point.
>
>
> |                                    | Unwatermarked | Unicode ICW | Initials ICW | Lexical ICW | Acrostics ICW |
> |------------------------------------|---------------|-------------|--------------|-------------|---------------|
> | Percentage of identified watermark |      49%      |    0.510    |     0.780    |    0.358    |     0.550     |
> | ROC-AUC after removal              |       -       |    0.000    |     0.893    |    0.800    |     0.908     |

---

### Meta-Review · Area_Chair_vsNo · 2025-12-22

**Summary:**

This paper studies in-context watermarking for LLMs by embedding watermarks into LLM-generated text via prompt engineering.

Strengths:
(1) model-agnostic approach.
(2) overall comprehensive evaluation, with the trade-offs among several aspects including LLM requirements, detectability, robustness, and text quality.
(2) overall clear writing.

Weaknesses:
(1) lack of evaluation of the true positive rate at a very low false positive rate.
(2) lack of human evaluation.
(3) lack of evaluation of detectability of ICWs after a multi-turn conversation.
(4) improper position of the novelty of the proposed work in the context of the existing work (by one reviewer).
(5) practical limitation of the proposed work that limits the applicability of the proposed method (by one reviewer).

**Reviewer Concerns:**

(1) lack of evaluation of the true positive rate at a very low false positive rate was addressed.
(2) the concern regarding evaluating with one more LLM was addressed.
(3) the concern regarding the multi-turn conversation was also addressed.
(4) Lack of human evaluation is still outstanding.

**Reviewer Scores:**

The score (2) of reviewer PAYJ is unlikely to change (who explicitly mentioned authors did not address his/her concerns).
The remaining three reviewers are likely to keep their positive scores (6/6/8).

---

### Decision · Program_Chairs · 2026-01-26

Accept (Poster)